# Robust Canonicalization through Bootstrapped Data Re-Alignment

## Abstract

Fine-grained visual classification (FGVC) tasks, such as insect and bird identification, demand sensitivity to subtle visual cues while remaining robust to spatial transformations. A key challenge is handling geometric biases and noise, such as different orientations and scales of objects. Existing remedies rely on heavy data augmentation, which demands powerful models, or on equivariant architectures, which constrain expressivity and add cost. Canonicalization offers an alternative by shielding such biases from the downstream model. In practice, such functions are often obtained using canonicalization priors, which assume aligned training data. Unfortunately, real-world datasets never fulfill this assumption, causing the obtained canonicalizer to be brittle. We propose a bootstrapping algorithm that iteratively re-aligns training samples by progressively reducing variance and recovering the alignment assumption. We establish convergence guarantees under mild conditions for arbitrary compact groups, and show on four FGVC benchmarks that our method consistently outperforms equivariant, and canonicalization baselines while performing on par with augmentation.

## 1 Introduction

Fine-grained visual classification (FGVC) systems often learn spurious correlations with nuisance factors such as scale and rotation—a form of learning bias that undermines model reliability. We study this problem in biodiversity monitoring, where insect and bird identification systems must remain robust despite systematic imaging biases. Insects are typically photographed top-down under arbitrary in-plane rotations and rescalings, while birds in the wild exhibit similar symmetries[1] Rather than learning invariance to these transformations, models often exploit them as shortcuts, creating a bias that degrades performance on out-of-distribution examples [8]. In this paper, we focus on two affine subgroups—rotation and scale (excluding translation, as residual localization errors after detection are usually minor)—and investigate methods to mitigate the induced learning biases.

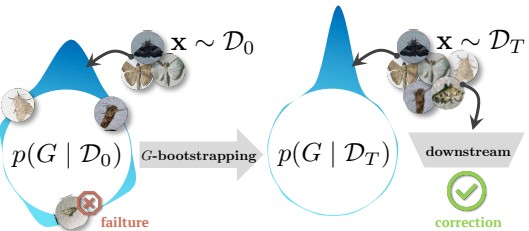

Figure 1: Our proposed $G$-bootstrapping aligns dataset $\mathcal{D}$ gradually (over $T$ time steps) during training by minimizing the variance over a specified compact group $G$. In the above example, $G$ is the group of rotations and $p(G \mid \mathcal{D})$ represents the distribution of angles in $\mathcal{D}$. The process runs jointly with the training of a canonicalizer, which aligns samples during inference — shielding geometric noise from the downstream model.

---

[1]Different scales arise from natural size variation across individuals and the proximity of the observer to the animal; and captures of flying birds come in arbitrary orientations.

Existing approaches to spatial robustness follow three main directions. The most widely used is *data augmentation*, where semantically redundant samples are added to the training set to encourage invariance [11]. This forces models to stack transformed replicates of the same filter [23] or average fuzzy per-class features across transformations [2, 5], which consumes model capacity and can introduce feature biases [1]. Augmentation does not provide mathematical guarantees, an issue that becomes critical for generalization and out-of-distribution prediction, where such models fall short [15], in contrast to *equivariant models*, which enforce explicit inductive biases on the hypothesis space. Vanilla convolutional neural networks (CNNs) are translation-equivariant [31], but remain sensitive to rotations and scale. Rotation and scale equivariance can be achieved by lifting filters to group-parameterized filter banks [3] or by using steerable basis functions such as circular harmonics [4, 30]. However, equivariant and invariant[2] architectures trade off flexibility and efficiency: restricting the filter space reduces expressivity, while explicit group representations increase memory and computational cost.

A third line of work is *canonicalization*, which seeks to transform inputs into a canonical form by "undoing" transformations. Recent advances include canonicalizers based on equivariant scoring functions [13, 22], and pseudo-canonicalizers trained on finite datasets [12, 26, 28] or obtained via test-time computations [27]. These methods offer flexible solutions for spatial robustness without constraining downstream architectures. The most popular approaches to learning canonicalizers are based on optimizing a scoring function [13], which assumes globally aligned datasets. In practice, however, this assumption is almost never satisfied: real-world FGVC datasets exhibit heterogeneous rotations and scales, leading canonicalizers to either overfit spurious orientations or collapse to averaged, non-meaningful canonical forms.

**Contributions**   To address this challenge, we propose a simple yet effective *bootstrapping scheme* that progressively re-aligns training data. By iteratively correcting high-loss samples toward a canonical pose, our method reduces spatial variance without constraining downstream architectures or requiring expensive test-time computations. We show that under mild assumptions, our algorithm contracts the spatial variance of the dataset with exponential convergence. We demonstrate improved spatial robustness across two FGVC benchmarks, validating the importance of dataset re-alignment for biodiversity monitoring and related domains.

## 2   Canonicalization

Groups formalize symmetries in data: for example, the special orthogonal group $SO(2)$ represents planar rotations, where each element corresponds to a rotation by an angle $\theta \in [0, 2\pi)$. A *representation* of a group $G$ is a homomorphism $\rho : G \to \mathrm{GL}(\mathcal{X})$ that maps each group element $g \in G$ to a linear transformation $\rho(g)$ on a vector space $\mathcal{X}$, such as a $3 \times 3$ rotation matrix acting on homogeneous image coordinates. In appendix A we provide an introduction to the fundamentals in group theory for readers unfamiliar with it. Canonicalisation is the process of assigning to each orbit of a group action a single, standard representative, called its *canonical form*. In this paper, we focus on orbit or contractive canonicalisers $\phi$ [21], which aim to "quotient out" group symmetries. They collapse the $G$-transformed input space $\mathcal{X}/G$ onto a subspace consisting of canonical representatives. That is, $\phi$ selects one representative from each orbit and returns it consistently.

**Definition 2.1** (Orbit Canonicalizer). *Let a group $G$ act on a vector space $\mathcal{X}$. A function $\phi : \mathcal{X} \to \rho(G)$ is called an* orbit canonicalizer *if for all $\mathbf{x} \in \mathcal{X}$ and $g \in G$, $\phi(\rho(g)\mathbf{x}) = \rho(g)$.*

**Definition 2.2** (Canonicalized Classifier). *Using a canonicalization function $\phi$, a classifier $\mathcal{X} \to \mathcal{Y}$ expressed by a parameterized distribution $p(y \mid \mathbf{x})$ can be canonicalized, such that $p_\phi(y \mid \mathbf{x}) := p(y \mid \phi(\mathbf{x})^{-1}\mathbf{x}) = p(y \mid \phi(\rho(g)\mathbf{x})^{-1}\rho(g)\mathbf{x})$ for all $g \in G, \mathbf{x} \in \mathcal{X}$.*

Consider the (Lie) group of continuous planar rotations called the special orthogonal group $SO(2)$. When $\phi$ predicts the rotation angle of $\rho(g)\mathbf{x}$, then its inverse $\rho(g)^{-1}$ maps back to its canonical form, i.e., the unbiased (upright) image $\mathbf{x}$. This collapses the continuous family of rotated copies of an image back into a single representative $\mathbf{x}$.

Canonicalization functions require $G$-equivariance of either the canonicalization function itself or an associated scoring function [13]. Let $s : \rho(G) \times \mathcal{X} \to \mathbb{R}$ be a scoring function modeled by a neural

---

[2]Equivariance is converted to invariance by pooling over the group dimension [3].

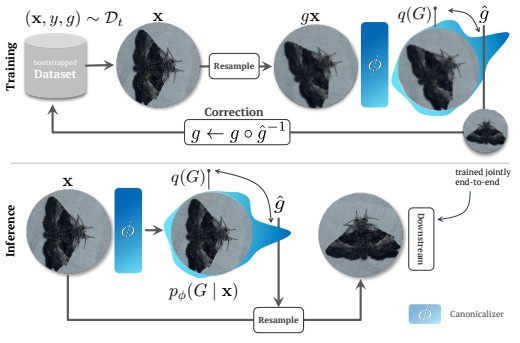

Figure 2: During training and inference, the distance between the prior $q(G)$ and the posterior $p_\phi(G \mid \mathbf{x})$ is leveraged to compute $\hat{g}$, which is used to correct the orientation of $\mathbf{x}$. This is used to bootstrap the training data over time and gradually align it to the prior.

**Algorithm 1:** Bootstrapping $G$-Alignment

**Require:** Training dataset $\mathcal{D}$, update interval $N$, update fraction $\alpha \in (0, 1]$
**Ensure:** Progressively $G$-aligned dataset $\mathcal{D}_T$
1: Set initial dataset state $\mathcal{D}_0 = \mathcal{D}, t = 0$
2: **for** epoch $e = 1, 2, \ldots$ **do**
3:     Train $\phi$ on $\mathcal{D}_t$ by $\mathcal{L} = \mathcal{L}_{\text{NLL}} + \lambda\mathcal{L}_{\text{prior}}$
4:     **if** $e \bmod N = 0$ **then**
5:         Compute $\hat{g}, \forall \mathbf{x} \in \mathcal{D}_t$ eq. (1)
6:         Compute $\mathcal{L}(\mathbf{x}), \forall \mathbf{x} \in \mathcal{D}_t$
7:         Select top-$\alpha|\mathcal{D}|$ samples $\widetilde{\mathcal{D}}_t \subset \mathcal{D}_t$
8:         Update $\mathbf{x} \leftarrow \rho(\hat{g})^{-1}\mathbf{x}, \forall \mathbf{x} \in \widetilde{\mathcal{D}}_t$
9:         Increment time step $t \leftarrow t + 1$
10:    **end if**
11: **end for**
12: **return** $\mathcal{D}_T$

network, which learns to minimize the energy [16] of the group element (e.g., the rotation angle) that acts on $\mathbf{x}$. With it, we can obtain a set of possible canonicalizers by

$$\hat{g} \leftarrow \phi(\mathbf{x}) \in \underset{\rho(g) \in \rho(G)}{\text{argmin}} \ s(\rho(g), \mathbf{x}). \tag{1}$$

The resulting canonicalizer satisfies definition 2.1 if $s$ is $G$-equivariant.[3] Self-symmetries (non-trivial stabilisers) imply that the minimizer may not be unique; in such cases, $\phi(\mathbf{x})$ selects an arbitrary element of the coset of minimizers. The scoring function $s$ parameterizes an energy function over the group orbit, which implicitly defines a distribution

$$p_\phi(G \mid \mathbf{x}) \quad \text{with} \quad \mathbb{P}_\phi(g \mid \mathbf{x}) \propto \exp(-s(\rho(g), \mathbf{x})). \tag{2}$$

This can be learned by fitting a canonicalization prior over $G$ in form of a Dirac distribution $q_{\mathcal{D}}$ [22]

$$\mathcal{L}_{\text{prior}} = \mathbb{E}_{\mathbf{x} \in \mathcal{D}} \left[ D_{\text{KL}} \left( q_{\mathcal{D}} \| p_\phi(\mathbf{x}) \right) \right] = -\mathbb{E}_{\mathbf{x} \in \mathcal{D}} \left[ \mathbb{E}_{p_\phi} \log q_{\mathcal{D}} \right] + \text{const.} \tag{3}$$

This is optimized as a regularizer on top of the downstream loss (like the negative log-likelihood). An $G$-equivariant neural network modeling $s$ is essential, as otherwise $p_\phi(\mathbf{x})$ would collapse to the constant prior $q_{\mathcal{D}}$ by overfitting. Without an equivariant backbone, the collapse can be prevented by an auxiliary loss, that enables approximate orthogonality of each representation along the orbit [24].

**The Need for $G$-Aligned Datasets** Implicit in (3) is the assumption that all samples in $\mathcal{D}$ are globally $G$-aligned. In practice, this assumption is rarely satisfied: real-world datasets exhibit heterogeneous affine perturbations. When trained on such data, $\phi$ interprets deviations from the dominant pose as noise. If the fraction of such deviations is small, $\phi$ may overfit by memorizing them; if large, it may underfit, failing to learn any consistent alignment. This brittleness arises from the mismatch between the unimodal prior $q_{\mathcal{D}}$ and the true orientation distribution $p_{\mathcal{D}}(G)$. The KL-regularization in eq. (3) enforces posterior consistency with $q_{\mathcal{D}}$, but when $p_{\mathcal{D}}(G)$ is multimodal or high-variance, no unimodal posterior can faithfully represent it.

## 3   Bootstrapping $G$-Aligned Datasets

We address this challenge by introducing a bootstrapping scheme that progressively re-aligns the dataset toward a unimodal canonical distribution on a compact group $G$. The core idea is iterative: we train the canonicalizer $\phi$ on the current dataset, identify the highest-loss samples as likely misaligned examples, realign these samples toward the canonical mode using eq. (3), and repeat the process. Over successive iterations, this procedure contracts the variance of poses in $\mathcal{D}$, yielding an increasingly

---

[3]That is, $s(\rho(g), \rho(h)\mathbf{x}) = s(\rho(h)^{-1}\rho(g), \mathbf{x})$ holds for all $g, h \in G$. The minimizing subset of eq. (1) also needs to be a coset of $\text{stab}_G(\mathbf{x})$ for all $\mathbf{x} \in \mathcal{X}$. For more detail and a formal proof see [13].

112   $G$-aligned dataset. Algorithm 1 provides the pseudo code and fig. 2 a visual overview of the training
113   and inference procedure.

114   Let $\mathcal{D}_t$ denote the dataset at step $t$ with $\mathcal{D}_0 := \mathcal{D}$. Bootstrapping modifies $\mathcal{D}_t$ by replacing a fraction
115   $\alpha \in [0, 1]$ of samples every $N \in \mathbb{N} \setminus \{0\}$ epochs with updated versions $\widetilde{\mathcal{D}}_t$[4], while retaining the
116   remaining subset $\widehat{\mathcal{D}}_t$, such that $\mathcal{D}_{t+1} = \widetilde{\mathcal{D}}_t \cup \widehat{\mathcal{D}}_t$ with preserved cardinalities $|\mathcal{D}_t| = |\mathcal{D}_{t+1}|$ for all $t$.
117   Let $p_{\mathcal{D}_t}(G)$ denote the distribution over $G$ at time step $t$. An update step $t \to t + 1$ is defined by the
118   linear interpolation resulting in the mixture distribution

$$p_{\mathcal{D}_{t+1}}(G) = (1 - \alpha)\, p_{\widehat{\mathcal{D}}_t}(G) + \alpha\, p_{\widetilde{\mathcal{D}}_t}(G). \tag{4}$$

119   Our bootstrapping procedure can be understood as a variance–contracting operator on compact
120   groups. Intuitively, each iteration reduces the variance of $p_{\mathcal{D}}(G)$ by re-aligning high-loss samples
121   toward a canonical pose, while leaving already well-aligned samples largely unaffected. We prove
122   that this contraction property guaranties exponential convergence of the re-alignment process under
123   mild assumptions. These results imply that the procedure is both stable and convergent (progressively
124   sharper canonical alignment over iterations).

125   Denote the Fréchet mean and variance by $\mu_t$ and $\sigma_t^2$, respectively. If the canonicalization procedure
126   is symmetric and unbiased, the mean drift becomes negligible, and curvature effects are small for the
127   affine group. Under the bootstrapping update with step size $\alpha$, the variance evolves as

$$\sigma_{t+1}^2 \approx (1 - \alpha)\sigma_t^2 + \alpha\widetilde{\sigma}_t^2, \tag{5}$$

128   which guaranties monotone contraction whenever $\widetilde{\sigma}_t^2 < \sigma_t^2$. Introducing a relative improvement
129   factor $\beta \in (0, 1)$ such that $\widetilde{\sigma}_t^2 = \beta\sigma_t^2$, we obtain exponential convergence of the variance,

$$\sigma_t^2 = \lambda^t \sigma_0^2, \qquad \lambda = 1 - \alpha(1 - \beta) \in (0, 1), \tag{6}$$

130   demonstrating that repeated canonicalization steps progressively concentrate the dataset around its
131   Fréchet mean. This ensures stable exponential convergence while exploiting the full contraction
132   effect. We provide a formal treatment in appendix B.

## 4   Experiments

134   We use a pretrained (ImageNet1k [6]) Swin-Base [19] for classification, which is trained end-to-end
135   with the canonicalizer $\phi$. We leverage a custom $G$-equivariant ResNet18 [10] family with a reduced
136   number of base latent dimensions of 32 to reduce the memory footprint. Using a $C_4$ filter bank as
137   in [3], these ResNets gain $C_4$-equivariance (discrete rotations). Using circular harmonics, we can
138   construct a steerable filter basis as in [4] to gain $SO(2)$-equivariance (continuous rotations). Using
139   a linear combination of Gaussian derivative basis filters as in [30], $\mathbb{R}^2 \times SO(2)$-equivariance is
140   obtained (i.e., rotoscale-equivariance with 8 angles and 8 scale factors). More details can be found in
141   appendix C and our publicly available source code. We focus our experiments on FGVC benchmarks
142   on insects and birds (for biodiversity monitoring systems). Specifically, we evaluate our method on
143   two FGVC benchmark datasets: EU-Moths [14] with 1650 samples and 200 classes, and NABirds
144   [29] with 23929 samples and 555 classes.

145   **Update Interval and Fraction**   We investigate how bootstrapping hyper-parameters affect down-
146   stream performance and convergence. Our method (algorithm 1) introduces two tunable parameters:
147   the update fraction $\alpha$ and the update interval $N$. Figure 3 shows performance on the vanilla test set
148   over 100 training epochs. The backbone's equivariance type—discrete or continuous—significantly
149   influences results. GCNNs [3] implementing discrete $C_4$-equivariance can only update samples by
150   $90°$ rotation multiples. Frequent updates with this constraint disrupt training and degrade perfor-
151   mance. In contrast, steerable CNNs [4] with continuous $SO(2)$-equivariance handle frequent updates
152   better due to gradual transformations. We found $N = 5$ and $\alpha = 1\%$ to yield optimal results overall.
153   During training on NABirds with steerable CNNs, only $1.2\%$ of samples were transformed, which
154   indicates that our bootstrapping focuses primarily on a few high-loss samples.

155   Multiple hyperparameter configurations of our bootstrapping algorithm outperform the canonicaliza-
156   tion prior (CanPrior) baseline on both vanilla and augmented test sets. Performance on the vanilla test

---

[4]We maintain transformation matrices along the images $\mathcal{D}$ and compose these matrices over time. This
minimizes the resampling error, as interpolation is applied only once as in [18].

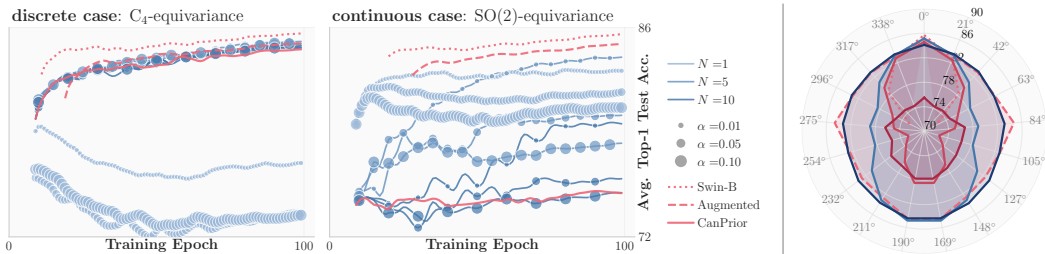

Figure 3: *(left)* Average top-1 test accuracy progression over 100 training epochs on NaBirds [29] using a $C_4$- [3] and a $SO(2)$-equivariant [4] backbone. *(right)* Average top-1 accuracy per angle on the rotation-augmented test set using $C_4$ [3] (with • / without •) and $SO(2)$ [4] (with • / without •).

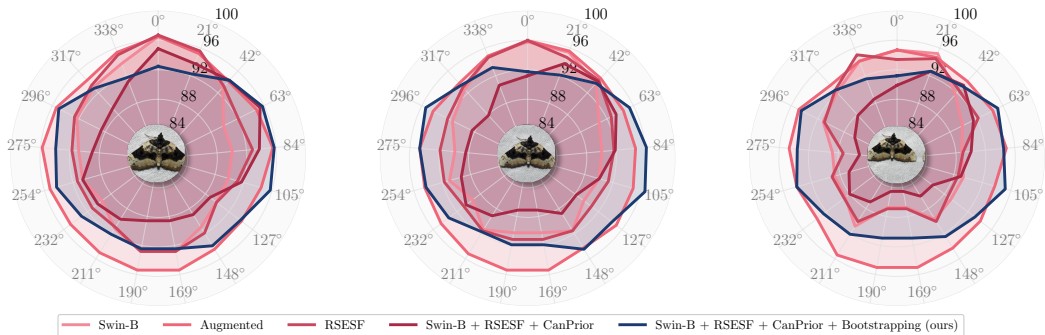

Figure 4: Average top-1 test accuracy over three rotoscale cosets (constant scale per subplot, $\{1, 1.125, 1.25\} \times C_{17}$) of EU-Moths [14] (goal: maximize area). A Swin-Base [19] is used as classifier, a RSESF [30] as a rotoscale equivariant backbone, a canonicalizer is trained using the canonicalization prior [22] with and with our bootstrapping.

set matches or slightly trails the vanilla and augmented baselines (see training progressions). However, on the rotation-augmented test set (fig. 3 right), performance significantly improves, matching the augmented baseline. To investigate robustness to rotation and scaling, we augment the test set with $C_{17}$ rotations (including $0°$) and scalings $1, 1.125, 1.25$ (identity and $12.5\%$, $25\%$ zoom-out). Figure 4 shows the results on EU-Moths [14], which contains significantly less geometric noise. Our bootstrapping procedure improves rotoscale robustness significantly compared to all baselines except the augmented one. Since augmented training employs the same transformations during testing, no distribution shift occurs, unlike in the other methods. Interestingly, performance of our method does not peak at the identity transformation. This indicates that the untransformed samples are nonetheless altered, resulting in a mean drift (see appendix B) and a diminished performance.

## 5   Conclusion

This paper addressed the problem of pose bias in fine-grained visual classification. We show that canonicalization functions trained on misaligned datasets are brittle, and propose a bootstrapping scheme that progressively re-aligns data to improve downstream robustness. Our theoretical analysis established variance contraction guarantees, and experiments confirmed strong empirical performance across multiple benchmarks.

**Limitations and Future Work**   Our method assumes that pose distributions are unimodal and can be contracted toward a canonical mode. Datasets with inherently multimodal or uniform poses may therefore challenge the bootstrapping scheme. Furthermore, although the computational overhead is modest, repeated updates may complicate large-scale training pipelines. Future directions include extending bootstrapping to multimodal orientation priors and integrating uncertainty estimates to extend beyond the rather simple highest-loss selection process of our algorithm. Instead of static update fractions and intervals, these might need to vary over time controlled by a scheduler to improve convergence. Beyond FGVC, our framework could be applied to broader domains where geometric bias affects recognition, such as medical imaging or remote sensing.

## A  Preliminaries in Group Theory

Consider the case of image classification, where $(\mathbf{x}, y) \sim \mathcal{D}$ denotes an image $\mathbf{x} \in \mathcal{X}$ and its label $y \in \mathcal{Y}$. For this paper, we focus on image data, where $\mathbf{x} \in \mathbb{R}^{C \times H \times W}$ denote a discrete image tensor obtained by sampling a continuous signal, where $H$ is the height, $W$ the width, and $C$ the number of channels (usually $C = 3$). Classification often comes with label-invariant transformations. For instance, an insect rotated by $90°$ is still the same insect. Formally, data points in $\mathcal{X}$ that share the same label are called *equivalence classes*. Many natural data transformations are symmetries that can be formalised as group actions. For example, the special orthogonal group $SO(2)$ describes planar rotations, where each element corresponds to a rotation by some angle $\theta \in [0, 2\pi)$. In general, a group $G$ acts on the vector space $\mathcal{X}$ via a homomorphism $\rho$ (called the group representation), so that each $g \in G$ is associated with a transformation $\rho(g) : G \to \mathcal{X}$. This can be a rotation matrix $\mathbf{R}_\theta \in \mathbb{R}^{3 \times 3}$ in homogeneous coordinates when $G = SO(2)$, or a translation matrix $\mathbf{T}_k \in \mathbb{R}^{1 \times 3}$ for a shift by $k$.

A classifier $f : \mathcal{X} \to \mathcal{Y}$ is said to be *equivariant* if $f(\rho(g)\mathbf{x}) = \rho'(g)f(\mathbf{x})$ for all $g \in G$, where $\rho$ and $\rho'$ denote the actions on input and output spaces. The special case where $\rho'(g)$ is the identity transform yields *invariance*. We define the *orbit* of $\mathbf{x}$ under $G$ as $G\mathbf{x} = \{g\mathbf{x} \mid g \in G\}$, i.e. the set of all transformed versions of $\mathbf{x}$. The collection of orbits forms the *orbit space* $\mathcal{X}/G = \{G\mathbf{x} \mid \mathbf{x} \in \mathcal{X}\}$, which partitions $\mathcal{X}$ into equivalence classes of geometrically indistinguishable objects. The *stabiliser* $\mathrm{stab}_G(\mathbf{x}) = \{g \in G \mid g\mathbf{x} = \mathbf{x}\}$ is the subgroup of elements leaving $\mathbf{x}$ unchanged. A *coset* of a subgroup $H \leq G$ is a set of the form $gH = \{gh \mid h \in H\}$ for some $g \in G$.

## B  Stability and Convergence

Let $d_G(\cdot, \cdot) : G \times G \to \mathbb{R}_+$ denote the geodesic distance induced by a bi-invariant Riemannian metric on $G$.[5] The Fréchet mean and variance of $\mathcal{D}_t$ are defined as

$$\mu_t = \operatorname*{argmin}_{g \in G} \mathbb{E}_{h \sim p_{\mathcal{D}_t}}[d_G(h, g)^2] \quad \text{and} \quad \sigma_t^2 = \mathbb{E}_{h \sim p_{\mathcal{D}_t}}[d_G(h, \mu_t)^2]. \tag{7}$$

These quantities generalize the Euclidean mean and variance (where $d(\cdot, \cdot) := \|\cdot\|_2$) to compact groups. Let $\widetilde{\mu}_t$, $\widehat{\mu}_t$ and $\widetilde{\sigma}_t^2$, $\widehat{\sigma}_t^2$ define the Fréchet mean and variance of the subsets of $\mathcal{D}$. We assume that all distributions involved belong to the same parametric family, which allows consistent computation of the Fréchet mean and variance and ensures that the canonicalization step reduces the variance within this family. This assumption simplifies the derivations while remaining applicable in practical settings where the orientation distribution is unimodal and smooth.

**Lemma B.1.** *Let $G$ be a compact Lie group with a bi-invariant Riemannian metric. Then the Fréchet variance of the mixture distribution in eq. (4) evolves as*

$$\sigma_{t+1}^2 = (1 - \alpha)\sigma_t^2 + \alpha\widetilde{\sigma}_t^2 + (1 - \alpha)d_G(\mu_t, \mu_{t+1})^2 + \alpha d_G(\widetilde{\mu}, \mu_{t+1})^2 + \mathcal{C}, \tag{8}$$

*where $\mathcal{C}$ is a curvature correction term.*

The proof can be found in appendix D.1.

The additional term $\mathcal{C}$ originates from the intrinsic curvature of the group manifold: geodesic distances cannot, in general, be decomposed linearly as in flat Euclidean space. Intuitively, this correction accounts for the fact that averages in curved spaces are not simply additive, and distances expand or contract depending on the local geometry of $G$. Meanwhile, the terms $(1 - \alpha)d_G(\mu_t, \mu_{t+1})^2 + \alpha d_G(\widetilde{\mu}, \mu_{t+1})^2$ capture the variance inflation arising from shifts in the mean location between steps — a phenomenon we refer to as *mean drift*. When mean drift is negligible, the variance progression simplifies considerably.

**Lemma B.2** (Mean Drift Under Symmetry). *Assume that the distribution $p_{\widehat{\mathcal{D}}_t}$ is symmetric about its Fréchet mean $\widehat{\mu}_t$, and that the canonicalization procedure produces realigned samples with distribution $p_{\widetilde{\mathcal{D}}_t}$ that is also symmetric about its mean $\widetilde{\mu}$. Then the mean drift terms satisfy:*

$$(1 - \alpha)d_G(\mu_t, \mu_{t+1})^2 + \alpha d_G(\widetilde{\mu}, \mu_{t+1})^2 \approx 0 \tag{9}$$

*when the canonicalization errors are unbiased.*

---

[5]A bi-invariant Riemannian metric is one that is invariant under both left- and right-multiplication by group elements, ensuring that geodesic distances are consistent with the group structure. Compact Lie groups such as $SO(2)$ admit such metrics.

226  The proof can be found in appendix D.2.

227  In practice, the magnitude of mean drift is governed by the canonicalizer's prediction error. Allowing
228  more training between updates (larger $N$) reduces this error, thereby controlling drift. If, however,
229  the encoder $\phi$ introduces systematic pose biases, the dataset may converge to a distribution centered
230  at a tilted mean that is inconsistent with the canonical prior.

231  For compact Lie groups such as $SO(2)$, the curvature is bounded and the correction $\mathcal{C}$ remains
232  uniformly small. In flat manifolds the correction vanishes entirely ($\mathcal{C} = 0$), recovering the classical
233  Euclidean variance decomposition. Thus, under symmetric canonicalization and unbiased updates,
234  both mean drift and curvature corrections become negligible after a few iterations, leaving variance
235  contraction as the dominant effect driving alignment.

236  **Lemma B.3.** *From lemma B.1 with lemma B.2 and a negligible curvature correction we have*

$$\sigma_{t+1}^2 \approx (1 - \alpha)\sigma_t^2 + \alpha\widetilde{\sigma}_t^2. \tag{10}$$

237  The proof follows trivially from lemma B.1 and lemma B.2.

238  The relevance of variance contraction is that it directly quantifies the alignment of the dataset: a low
239  Fréchet variance indicates that the poses in $\mathcal{D}_t$ are tightly concentrated around the canonical mode $\mu$.
240  Consequently, dataset alignment can be rephrased as minimizing $\sigma_t^2$. The sequence of lemmas above
241  shows that, once mean drift and curvature corrections are controlled, the update dynamics simplify
242  to a weighted averaging of variances, which guarantees a monotone contraction under suitable
243  conditions. This provides both a theoretical and practical justification for the bootstrapping scheme:
244  repeated canonicalization steps shrink dispersion until the dataset becomes sharply $G$-aligned.

245  **Theorem B.4** (Variance contraction and exponential convergence)**.** *Let $(\mathcal{D}_t)_{t\geq 0}$ be datasets with*
246  *Fréchet variances $\sigma_t^2 > 0$. Fix $\alpha \in (0, 1)$ and assume negligible mean drift from lemma B.2. Then,*
247  *we have a variance contraction $\sigma_{t+1}^2 < \sigma_t^2$, which is implied by $\widetilde{\sigma}_t^2 < \sigma_t^2$. This can be formulated by*
248  *a relative improvement $\beta \in (0, 1)$ such that $\widetilde{\sigma}_t^2 = \beta\,\sigma_t^2$, which becomes*

$$\sigma_{t+1}^2 = \lambda\,\sigma_t^2, \qquad \lambda := 1 - \alpha(1 - \beta) \in (0, 1). \tag{11}$$

249  *If this holds for all t, we have $\sigma_t^2 = \lambda^t\sigma_0^2$. Hence, variance contracts exponentially to zero with rate*
250  *$-\log \lambda$.*

251  The proof can be found in appendix D.3.

252  Theorem B.4 formalizes that the bootstrapping updates contract variance at a geometric rate. The
253  contraction factor $\beta$ represents the effectiveness of the canonicalizer in reducing dispersion on the up-
254  dated subset: for instance, $\beta = 0.5$ indicates that variance of corrected samples halves in expectation.
255  The global variance update then follows a simple linear recurrence, yielding exponential convergence.
256  Intuitively, the process resembles a weighted averaging scheme where each iteration progressively
257  shrinks the spread of orientations. Importantly, larger update intervals $N$ allow the encoder $\phi$ to
258  improve its orientation estimates, thereby reducing mean drift and permitting larger update fractions
259  $\alpha$ without destabilization. A practical schedule might therefore to begin conservatively — small
260  $\alpha$, large $N$ — for initially dispersed datasets, and then gradually increase $\alpha$ or decrease $N$ once $\phi$
261  achieves low orientation error. Such schedules will be explored in future work. This ensures stable
262  exponential convergence while exploiting the full contraction effect.

## C  Further Experiments and Implementation Details

264  **Implementation Details**    All our models and baselines are trained with AdamW [20] with a cosine
265  learning rate decay starting at either $1e - 3$ or $1e - 4$. Models are trained for $100$ epochs. We use
266  dropout ($0\%$ to $30\%$) on the final linear layer, weight decay ($0$ to $1e - 3$), and mild augmentation.
267  With a $10\%$ chance images are blurred by a Gaussian kernel and/or the sharpness is increased. All
268  images are resized to $224 \times 224$px and a circular mask is applied (with zero-padding) to prevent
269  rotation artefacts to bias the model. We use Nvidia A40s for training and evaluation.

270  **Construction of Testsets**    Let $\mathcal{D}_{\text{test}} \subset \mathcal{D}$ denote the test set. To evaluate the robustness of our
271  canonicalized classifier $p_\phi$, we introduce two additional test sets. Both represent augmented variants
272  of $\mathcal{D}_{\text{test}}$. We construct the two orbit spaces $\mathcal{D}_{\text{test}}/\text{SO}(2)$ and $\mathcal{D}_{\text{test}}/(\text{SO}(2) \times \mathbb{R}^2)$. This holds all

273 rotation and roto-scaled variants of the original images in $\mathcal{D}_{\text{test}}$, respectively. In our experiments,
274 we use a finite version of this orbit space using a subgroup of both $\text{SO}(2)$ and $\mathbb{R}^2$ of order 17.
275 This ensures that the identity is present in the augmented datasets, while the transformations apply
276 symmetrically (like zooming in and out with equal amounts).

277 As we augment images by $G = \text{SO}(2) \times \mathbb{R}^2$, we expect only trivial stabilizers, hence low "symmetry
278 richness" in general. This results in orbit sizes $|G\mathbf{x}| \approx |G|$, $\forall \mathbf{x} \in \mathcal{D}_{\text{test}}$ according to the Orbit-
279 stabilizer theorem [7]. Hence, we expect a large test-time performance drop of models unable to cope
280 with these transformations.

## D Proofs

### D.1 Proof of lemma B.1

283 *Proof.* By definition, the Fréchet variance of the mixture is

$$\sigma_{t+1}^2 = \mathbb{E}_{g \sim p_{\mathcal{D}_{t+1}}}[d_G(g, \mu_{t+1})^2]. \tag{12}$$

284 Expanding over the mixture components gives

$$\sigma_{t+1}^2 = (1-\alpha)\mathbb{E}_{g \sim p_{\widehat{\mathcal{D}}_t}}[d_G(g, \mu_{t+1})^2] + \alpha\mathbb{E}_{g \sim p_{\widetilde{\mathcal{D}}_t}}[d_G(g, \mu_{t+1})^2]. \tag{13}$$

285 For each component, we apply the fundamental distance decomposition on Riemannian manifolds.
286 Working in the tangent space at $\mu_t$, let $v_g = \exp_{\mu_t}^{-1}(g)$ and $w = \exp_{\mu_t}^{-1}(\mu_{t+1})$. Using the exponential
287 map expansion and parallel transport properties on compact Lie groups [9, 17], the geodesic distance
288 satisfies

$$d_G(g, \mu_{t+1})^2 = d_G(g, \mu_t)^2 + d_G(\mu_t, \mu_{t+1})^2 - 2\langle v_g, w \rangle + R(g, \mu_t, \mu_{t+1}) \tag{14}$$

289 where $R(g, \mu_t, \mu_{t+1})$ represents curvature-dependent correction terms arising from the non-linearity
290 of the exponential map.

291 Taking expectation over $g \sim p_{\widehat{\mathcal{D}}_t}$ and applying the defining property of the Fréchet mean, we obtain

$$\mathbb{E}_{g \sim p_{\widehat{\mathcal{D}}_t}}[d_G(g, \mu_{t+1})^2] = \mathbb{E}[d_G(g, \mu_t)^2] + d_G(\mu_t, \mu_{t+1})^2 - 2\langle \mathbb{E}[v_g], w \rangle + \mathbb{E}[R(g, \mu_t, \mu_{t+1})] \tag{15}$$

$$= \sigma_t^2 + d_G(\mu_t, \mu_{t+1})^2 + \mathcal{C}_1, \tag{16}$$

292 where the cross-term vanishes because $\mathbb{E}[v_g] = \mathbb{E}[\exp_{\mu_t}^{-1}(g)] = 0$ by the first-order optimality
293 condition for the Fréchet mean (i.e., $\mathbb{E}\left[\langle \nabla d_G(g, \mu_t)^2, v_g \rangle\right] = 0$) [25, Theorem 2], and $\mathcal{C}_1 =$
294 $\mathbb{E}[R(g, \mu_t, \mu_{t+1})]$ collects the curvature correction terms.

295 Similarly, for the realigned component with Fréchet mean $\widetilde{\mu}$, applying the same distance decomposi-
296 tion in the tangent space at $\mu_t$ yields

$$\mathbb{E}_{g \sim p_{\widetilde{\mathcal{D}}_t}}[d_G(g, \mu_{t+1})^2] = \widetilde{\sigma}_t^2 + d_G(\widetilde{\mu}, \mu_{t+1})^2 + \mathcal{C}_2, \tag{17}$$

297 where $\mathcal{C}_2$ represents the corresponding curvature correction terms for the realigned subset.

298 Substituting eq. (15) and eq. (17) into eq. (13) gives

$$\sigma_{t+1}^2 = (1-\alpha)\left[\sigma_t^2 + d_G(\mu_t, \mu_{t+1})^2 + \mathcal{C}_1\right] + \alpha\left[\widetilde{\sigma}_t^2 + d_G(\widetilde{\mu}, \mu_{t+1})^2 + \mathcal{C}_2\right] \tag{18}$$

$$= (1-\alpha)\sigma_t^2 + \alpha\widetilde{\sigma}_t^2 + (1-\alpha)d_G(\mu_t, \mu_{t+1})^2 + \alpha d_G(\widetilde{\mu}, \mu_{t+1})^2 + \mathcal{C} \tag{19}$$

299 where $\mathcal{C} = (1-\alpha)\mathcal{C}_1 + \alpha\mathcal{C}_2$ collects all curvature correction terms. This completes the proof. $\quad\square$

### D.2 Proof of lemma B.2

301 *Proof.* By assumption, $p_{\widehat{\mathcal{D}}_t}$ is symmetric about $\mu_t$, and $p_{\widetilde{\mathcal{D}}_t}$ is symmetric about $\widetilde{\mu}$. Furthermore, the
302 canonicalization procedure applies unbiased corrections to the high-loss samples: for every sample
303 rotated by $g \in G$ in one direction, a corresponding sample rotated by $g^{-1}$ is corrected symmetrically.

304 From the properties of symmetric distributions on Lie groups [25], the Fréchet mean of a symmetric
305 distribution coincides with its center of symmetry. Therefore, we have $\widehat{\mu}_t = \mu_t$ and $\widetilde{\mu} = \mu_t$.
306 Consequently, the mean of the mixture satisfies $\mu_{t+1} = (1-\alpha)\mu_t + \alpha\widetilde{\mu} = \mu_t$, so that

$$d_G(\mu_t, \mu_{t+1})^2 = d_G(\widetilde{\mu}, \mu_{t+1})^2 = 0. \tag{20}$$

307 This shows that the mean drift terms in lemma B.1 vanish exactly under symmetric canonicalization.
308 Hence, under unbiased corrections, the mixture variance evolves solely according to the weighted
309 combination of component variances, and no systematic drift of the Fréchet mean occurs. $\square$

## D.3 Proof of theorem B.4

311 *Proof.* From lemma B.3 we have

$$\sigma_{t+1}^2 = (1-\alpha)\sigma_t^2 + \alpha\widetilde{\sigma}_t^2. \tag{21}$$

312 Subtracting $\sigma_t^2$ from both sides gives

$$\sigma_{t+1}^2 - \sigma_t^2 = \alpha(\widetilde{\sigma}_t^2 - \sigma_t^2), \tag{22}$$

313 Since $\alpha > 0$, the sign of $\sigma_{t+1}^2 - \sigma_t^2$ equals the sign of $\widetilde{\sigma}_t^2 - \sigma_t^2$. Therefore $\sigma_{t+1}^2 < \sigma_t^2$ iff $\widetilde{\sigma}_t^2 < \sigma_t^2$.
314 This inequality can be expressed by substituting $\widetilde{\sigma}_t^2 = \beta\sigma_t^2$ with $\beta \in (0,1)$. This results in linear
315 recurrence

$$\sigma_{t+1}^2 - \sigma_t^2 = \alpha(\widetilde{\sigma}_t^2 - \sigma_t^2) \tag{23}$$

$$\sigma_{t+1}^2 - \sigma_t^2 = \alpha(\beta\sigma_t^2 - \sigma_t^2) \tag{24}$$

$$\sigma_{t+1}^2 = \sigma_t^2\alpha(\beta - 1) + \sigma_t^2 \tag{25}$$

$$= \sigma_t^2(\alpha(\beta - 1) + 1). \tag{26}$$

$$\tag{27}$$

316 We can rewrite $(\alpha(\beta - 1) + 1) = 1 - \alpha(1 - \beta)$, which is the linear recurrence term $\lambda$ in theorem B.4.
317 If this holds for all time steps $t$, then $\sigma_{t+1}^2 = \lambda^t\sigma_t^2$. This gives a contraction rate of $-\log \lambda$ until zero
318 variance is reached. This concludes the proof. $\square$

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
