# OpenReview forum: "Robust Canonicalization through Bootstrapped Data Re-Alignment"
_EurIPS.cc/2025/Workshop/UPLB — UPLB2025_

### Official Review · Reviewer_agc8 · 2025-10-17
**Assessment**

**Rating:** 7
**Confidence:** 2

**Review:**

This work proposes a method to align images which are related by symmetries (e.g. rotations) in classification tasks through *canonicalization*: a method that to fix a representative element to every point in the of a given group. Based on this notion, they propose a bootstrap algorithm for canonicalization of data, focusing on rotations and rescalings, and provide both theoretical guarantees and numerical results.

The paper fits the workshop theme in the sense that certain classes of images are typically photographed in a biased way, such as insect pictures which are taken from the top. Therefore, canonicalization can be seen as a form of geometrical bias reduction.

---

### Decision · Program_Chairs · 2025-11-03

Accept (Poster)